Three new Luticola D.G.Mann (Bacillariophyta) species from Rapa Nui (Easter Island) found in terrestrial diatom assemblages dominated by widely distributed taxa

Peszek Łukasz lpeszek@ur.edu.pl 1
Rybak Mateusz 1
Lange-Bertalot Horst 2
Kociolek John Patrick 3
Witkowski Andrzej 4
1 Department of Agroecology, Institute of Agricultural Sciences, Land Management and Environmental Protection, University of Rzeszów , Rzeszów , Poland
2 Institute of Ecology, Evolution and Diversity, Goethe University , Frankfurt am Main , Germany
3 Museum of Natural History and Department of Ecology and Evolutionary Biology, University of Colorado , Boulder , CO , United States of America
4 Institute of Marine and Environmental Sciences, University of Szczecin , Szczecin , Poland
Sosa Victoria
Electronic publication date: 2021 Apr 5
Publication date: 2021
Volume: 9
Electronic Location ID: e11142
Received 2020 Nov 23; Accepted 2021 Mar 2
Copyright: ©2021 Peszek et al.
Copyright year: 2021
Copyright holder: Peszek et al.
License: This is an open access article distributed under the terms of the Creative Commons Attribution License, which permits unrestricted use, distribution, reproduction and adaptation in any medium and for any purpose provided that it is properly attributed. For attribution, the original author(s), title, publication source (PeerJ) and either DOI or URL of the article must be cited.
License URL: https://creativecommons.org/licenses/by/4.0/

Keywords: SE Pacific, Oceanic Islands, Easter Island, Endemism, Aerophytic diatoms, Biogeography, Luticola, Diatoms, Bacillariophyceae

Funding: Polish Ministry of Science and Higher Education 026/RID/2018/19 The project was funded by the Polish Ministry of Science and Higher Education under the name of “Regional Excellence Initiative” in the years 2019–2022 Project No. 026/RID/2018/19. There was no additional external funding received for this study. The funders had no role in study design, data collection and analysis, decision to publish, or preparation of the manuscript.

==============================
Background

Rapa Nui (Easter Island = Isla de Pasqua) is of volcanic origin, best known for about 900 man-made stone statues known as moai. It is one of the most isolated inhabited islands on Earth and studies on the diatoms of Rapa Nui are very few.

Methods

Light (LM) and electron microscopic (SEM) observations of a single sample collected from Rapa Nui are presented. The samples (mix of soil and organic detritus) were collected from ground of cave entrance.

Results

The samples were characterized by low diatom diversity and strongly dominated by terrestrial (soil) forms. Among the taxa present in the material studied were cosmopolitan forms of the genera Humidophila, Nitzschia, Angusticopula, Orthoseira, Tryblionella and Luticola. Whereas most of taxa of the enumerated genera were identifiable, only one among four Luticola species distinguished in the samples studied was identified. This taxon was L. ectorii, a cosmopolitan species known previously from South America (Brazil) and Asia (China). The three remaining species could not be assigned to any established species. Therefore, based on external and internal morphological features from light and scanning electron microscopic analysis, we describe here three species new to science, including: L. georgzizkae, L. rapanuiensis and L. moaiorum. All three taxa new to science are compared to established Luticola species and their significance for the global distribution of the genus is discussed.

Introduction

Studies on the diatoms of Rapa Nui are very few (e.g., Cocquyt, 1991). These reports have documented species composition of the diatom assemblages from crater lakes and water pools near the coast. These studies used light microscopy to identify 70 taxa in freshwater lakes and brackish water pools near the coast. During these studies no new taxa were described and most of the identified taxa were considered as cosmopolitan.

From marine waters around Rapa Nui the new diatom genus and species Florella pascuensis Navarro was also described at water depths of 30–40 m and epiphytic on Halimeda sp. and Padina sp. (Navarro, 2002). Diatoms from a core taken in an inland lake have been studied along with other proxies by Dumont et al. (1998) to reconstruct an arrival of alien societies and their impact on the famous moai quarry (e.g.,  Hamilton, 2013).

Terrestrial habitats are places of characteristic diatom floras, dominated by resistant species which are able to thrive in conditions of limited water availability. Among such taxa are representatives of Luticola D.G. Mann (e.g.,  Round, Crawford & Mann, 1990; Levkov, Metzeltin & Pavlov, 2013). Luticola was established to accommodate species included in Naviculae sect. Punctatae. Luticola mutica (Kütz.) D.G.Mann was selected as the generitype. Luticola is one of the few diatom genera which earned an extensive treatise with LM and scanning electron microscopy (SEM) (Levkov, Metzeltin & Pavlov, 2013). These authors estimated the diversity of the genus to be ca. 200 taxa. Algaebase lists 239 species names in the database at present, as well as 23 infraspecific names. Of the species names, 236 have been flagged as accepted taxonomically on the basis of the listed literature under the species name (M.D. Guiry in Guiry & Guiry, 2020). The list shall be soon expanded with several new species very recently described from Western Ghats in India (Lokhande et al., 2020). The genus name was derived from Latin word “Lutum” meaning “mud” hence Luticola means mud dwelling (Round, Crawford & Mann, 1990). Indeed, many of the Luticola species, including the generitype L. mutica, are inhabitants of tidal mudflats and characterized by high tolerance of inorganic and organic pollutants therefore indicative of poor water quality (Lange-Bertalot, 1977; Lange-Bertalot et al., 2017). However, with recent discoveries and newly established taxa, some Luticola species may be characteristic of pristine lacustrine, swamp, riverine, estuarine, mangrove and terrestrial (soil) microhabitats (Levkov, Metzeltin & Pavlov, 2013; Van de Vijver, Frenot & Beyens, 2002; Ba̧k et al., 2017; Ba̧k et al., 2019; Chattová et al., 2017; Simonato et al., 2020; Lokhande et al., 2020).

Likewise, the biogeography of Luticola is interesting. Luticola taxa have been chiefly distributed in the Holarctic plant realm, especially in Eurasia (Navicula mutica Kützing) and North America (Hustedt, 1962; Patrick & Reimer, 1966; Krammer & Lange-Bertalot, 1988). A number of taxa have been described and known to occur or cited in the tropics (Hustedt, 1937–1939) or in high latitudes of the Southern Hemisphere (Heiden & Kolbe, 1928; Giffen, 1966; Cholnoky, 1954). A change in our understanding of the distribution of Luticola taxa is occurred in the mid-1990s and a wealth of Luticola have been described as new to science from tropical rainforests of South America (Metzeltin & Lange-Bertalot, 1998; Metzeltin & Lange-Bertalot, 2005; Bustos, Morales & Maidana, 2017; Straube, Tremarin & Ludwig, 2017; Da Silva-Lehmkuhl et al., 2019), in the Andes (Rumrich, Lange-Bertalot & Rumrich, 2000; Simonato et al., 2020) and in South-East Asia (e.g.,  Glushchenko & Kulikovskiy, 2015; Glushchenko, Kulikovskyi & Kociolek, 2018). High diversity of Luticola also has been observed in Antarctica and Sub-Antarctic Islands (Van de Vijver, Frenot & Beyens, 2002; Kopalová et al., 2011; Chattová et al., 2017; Kochman-Kedziora et al., 2020). The discussion on this biogeographic phenomenon has been recently published by Kociolek et al. (2017). In addition, Simonato et al. (2020) recognized different morphological groups in the genus, and noted their biogeographic distributions, including a small group of species differentiated from all others that are known only from South America.

The aim of our study was to analyze the species diversity in a set of three samples taken from terrestrial (cave) habitats from the Rapa Nui using high resolution LM and field emission SEM. Although only one sample turned out to be rich in diatoms with moderate species diversity, and many of the taxa present were cosmopolitan forms known from terrestrial and aerophytic habitats world-wide, with a few typical of South America. Of the Luticola species present in this sample, we recognized four species, three of which we describe here as new to science. In this way we confirm the special status of Luticola in the Southern Hemisphere and at Rapa Nui as an area of diversity for this genus.

Materials & Methods

Study area

Rapa Nui (Easter Island = Isla de Pasqua) is located in the southeastern Pacific Ocean and is a special territory of Chile located 3,580 km off the coast of this country. It is one of the most isolated inhabited islands on Earth. The island is 21 km wide, with an area of 166 km2. The climate on Rapa Nui is classified as subtropical marine. Annual mean temperature is 20.8 °C with an annual average daily variation of 6.3 °C. Annual mean precipitation recorded is nearly 1,217 mm (Quilliam et al., 2014). Because of its volcanic origin, Rapa Nui has a large number of caves (Ciszewski, Ryn & Szelerewicz, 2009). The porous nature of the lava, as well as the island’s caves, prevents the formation of watercourses, since rainwater is immediately absorbed into the ground. However, a few springs are found near the north coast. There are only three lakes where rainwater gathers: the craters of Rano Kau, Rano Raraku and Rano Aroi (Charola, 1994)

Rapa Nui is best known for about 900 man-made stone statues known as moai. The vascular plant flora of Rapa Nui is extremely poor compared to other tropical islands, reflecting its young geological age, small size, and high degree of isolation (Aldén, 1990). Nearly 90% of the territory is covered by herbaceous vegetation, with species of Poaceae (grasses); most of them are alien (Etienne & Faúndez, 1983; Finot et al., 2015). 23% of vascular flora is comprised of endemic species, and 20 species are native, 10 of which are endemic but have disappeared or are endangered (Dubois et al., 2013). Zizka (1991) identified 179 species of flowering plants in total. It seems that the original vegetation of the island was represented by palm-dominated forests but was replaced by a large number of introduced species that became naturalized (Rull et al., 2010). The flora of Rapa Nui is particularly poor due to the human activity, including deforestation, introduction of the Polynesian rat: (Rattus exulans Peale, 1848) fire and agriculture. Deforestation would have occurred about AD 1,000–1,200 or even 600 years later (Mann et al., 2008). Around AD 1650 social collapse of Rapa Nui occurred that was accompanied by warfare, a crash in population size, and cultural changes (Diamond, 2005). The history of island has been interpreted in different ways. One interpretation suggests an uncontrolled growth of the human population resulted in the destruction of the natural vegetation and led to near extinction of inhabitants (Diamond, 2005). A second theory suggests disease due to microorganisms could have been introduced by European arrivals (Hunt, 2007). A third scenario suggests introduction of the rapidly-reproducing rat may have done more damage to the native plants and animals than the Polynesians (Towns, Atkinson & Daugherty, 2006). A fourth interpretation of the disaster on Rapa Nui holds that environmental changes beyond the control of humans triggered the societal collapse (Diamond, 2005).

Figure 1 Study area.

(A) Location of Rapa Nui island (arrow). (B) Location of the studied cave.

Sampling

The samples were collected from cave number 436, on 30 April 1988 by Professor Dr. Georg Zizka during his work on flowering plants of Rapa Nui project (see Zizka, 1991) (Fig. 1). Samples in the form of soil and organic detritus were collected from the ground of the cave entrance. The sampling site was exposed to natural light. Samples composed of soil mixed with organic matter were boiled with 30% hydrogen peroxide (H2O2) for a few hours at 150 °C and then rinsed and settled 5 times with deionized water. The resulting cleaned diatom material was pipetted onto coverslips and dried, and then mounted on glass slides using Naphrax® mounting medium (Brunel Microscopes Ltd, Wiltshire, U.K.). LM observations of the cleaned material were made with a Zeiss Axio Imager A2 and Zeiss Axio Imager M2 (Carl Zeiss, Jena, Germany) using a × 100 Plan Apochromatic oil immersion objective (NA = 1.46) equipped with Differential Interference Contrast (DIC). Diatom images were captured with a Zeiss AxioCam ICc5 camera (Jena, Germany). 300 diatom valves were counted for to establish diatom species composition in the samples. For SEM examination, a few drops of cleaned material were put onto Whatman Nuclepore polycarbonate membrane filters (Fisher Scientific, Schwerte, Germany). Once dried, the membranes were mounted onto aluminum stubs and coated with 20 nm of gold. SEM observations were performed at the University of Rzeszów, using a Hitachi SEM SU8010. Diatom terminology follows Round, Crawford & Mann (1990) and Levkov, Metzeltin & Pavlov (2013).

Figure 2 Luticola georgzizkae sp. nov.

(A–N) LM images showing size diminution series. (A, B, E, G) Valve morphotypes with clearly visible marginal spines (arrows). (D) holotype specimen. (H) isotype specimen. (O–AC) SEM images. (O–R) External view of the valve face with spines (arrow in Fig. R). (S) External valve view of specimen without spines. (T) Detailed view of external proximal raphe endings associated with stigma (arrow). (U, V) Detailed view of external distal raphe endings terminating below apices (arrow in Fig. U). (W, X) Internal view of the whole valves. (Y) Internal view of central area and distal raphe endings terminating below the valve apices (arrow). (Z) Internal view of proximal raphe endings, note marginal channel (arrow), and stigma (black arrow). (AA) Girdle view of the whole frustule. (AB) Detailed valve mantle in girdle position with clearly visible marginal spines (arrow). (AC) Detached girdle band. Scale bars: (A–N, W) 10 µm, (O–S, X, Y, AA–AC) 5 µm, (T–V, Z) 3 µm.

Results

From three collected samples, one of them completely barren, the second one only with few diatom valves and the third was abundant in diatoms. In this particular sample besides the newly-described Luticola species represented by numerous well-established populations, we also identified 24 diatom species. The genus Luticola was represented in this small sample by four species, three of which are described herein as new to science and Luticola ectorii Levkov, Metzeltin & Pavlov (Levkov, Metzeltin & Pavlov, 2013).

Novel taxa –diagnosis

Luticola georgzizkae sp. nov. Witkowski, Lange-Bertalot, M. Rybak & Peszek (Fig. 2).

Light microscopy: Valves rhombic to rhombic-lanceolate in larger specimens, with slightly undulate margins (Figs. 2A–2N), usually bearing a row of distinct spines (Figs. 2B, 2E and 2G). Valve length 8.3–35.2 µm, width 6.4–10.3 µm. Axial area narrow and linear. Central area rectangular bordered by single row of areolae. Transapical striae clearly punctate, radiate, 16–19 in 10 µm, contain 3–4 areolae per stria.

Scanning electron microscopy: Externally, raphe linear not reaching the valve apices, especially in large specimens, with external proximal endings slightly deflected to the side opposite the stigma (Figs. 2O–2T and 2V), distal raphe endings straight to curved slightly in the same direction as the proximal ends (Figs. 2O–2S, 2U and 2V). Areolae rounded to transapically elongate, radiate (Figs. 2T and 2V). Transapically elongated stigma located close to valve margin in the expanded central area (Figs. 2S, 2T and 2V). The external surface of the valve makes it possible to distinguish two morphotypes. First with large marginal spines, small spines on whole valve surface, deeply positioned areolae and shorter raphe (Figs. 2O–2R). The second, without marginal spines and with smooth, flat valve surface and raphe slit which almost reach the valve apices (Fig. 2S).

Internally, raphe branches simple and straight, proximal ends slightly deflected to stigma side (Figs. 2W–2Z). Small stigma positioned close to the valve margin with a circular and lipped structure without a distinct opening (Fig. 2Z). Areolae variable in shape, single row of areolae positioned also on valve mantle (Figs. 2Y and 2Z). Areolae occluded by hymenes, near the central area separated, becoming a continuous strip near valve apices (Figs. 2X and 2Y). Copulae opened with perforations and covered by small silica granules (Fig. 2AC). Marginal channel tube-like, located on valve face/mantle junction, internally occluded with hymenes (Fig. 2Z), widened in the central valve part, on the side opposite to stigma (Fig. 2X).

For morphometric description 30 valves were analyzed under LM, and 15 valves under SEM.

Type locality: Rapa Nui, cave interior, Coordinates: 27°6′42″S 109°24′14″W, collected on 30.04.1988 by Prof. Georg Zizka from Senckenberg Institute, Frankfurt am Main, Germany.

Holotype: holotype designated here: slide SZCZ 20608 (Fig. 2D) in the collection of Andrzej Witkowski at the Institute of Marine and Environmental Sciences, University of Szczecin.

Isotype: slide number 2020/11 in the Diatom Collection of the Institute of Agricultural Sciences, Land Management and Environmental Protection at the University of Rzeszów.

Etymology: The new species is named in honor of Professor Dr. Georg Zizka for his scientific achievements in Botany.

Luticola rapanuiensis sp. nov. M.Rybak, Peszek, Witkowski & Lange-Bertalot (Fig. 3).

Light microscopy: Valves rhombic to elliptical with obtusely rounded apices. Valve length 5.2–15.9 µm, width 4.3–7.6 µm. Axial area narrow and linear, central area bow-tie shaped and bordered by 3–4 small areolae. Transapical striae radiate, 16–19 in 10 µm, clearly punctate, composed of 2–3(4) areola per stria (Figs. 3A–3P).

Figure 3 Luticola rapanuiensis sp. nov.

(A–P) LM images showing size diminution series. (F) Isotype specimen. (H) Holotype specimen. (Q–AB) SEM images. (Q–V) External views of the entire valve. (T) External valve view, with clearly visible ghost areola (arrow). (U) External valve view of central raphe endings with elongated shape and grooves (arrow). (W) Close up of the specimen illustrated in Fig. T, with large ghost areola (arrow), stigma (black arrow), proximal and distal raphe endings slightly bent to the same side. (X) Close up of central part of the valve, with proximal raphe endings evident. (Y) Close up of the valve apex with siliceous ridges on valve face/mantle conjunction (arrow) clearly evident. (Z) Valve in a girdle view. (AA) Internal view of whole valve. (AB) Internal view of valve illustrating stigma (arrow) and central raphe endings. Scale bar: (A–P) 10 µm, (Q-U, X, Z, AA) 5 µm, (V, Y) 4 µm, (W) 3 µm, (AB) 1 µm.

Scanning electron microscopy: Externally, raphe straight to slightly bent, contained within a narrow axial area (Figs. 3Q–3V). External proximal raphe ends curved towards the stigma, which is small (Fig. 3W), and positioned near the margin in the expanded central area, continuing with irregular shallow grooves often conjoined with areolae which bordered central area (Fig. 3U). Ghost areolae often observed within the central area (Fig. 3W). Distal raphe ends are curved to tightly hooked in the same direction as the proximal ends (Figs. 3W and 3Y). Striae strongly radiate, composed of 2–4 rounded areolae (Figs. 3Q–3V). On valve face/mantle junction siliceous ridges are present (Figs. 3X and 3Y).

Internally, raphe simple and straight with proximal ends terminating in a small, internally-elevated central nodule; distal endings terminate simply (Figs. 3AA and 3AB). Areolae occluded by hymenes and not forming continuous strip across valve (Fig. 3AA). Stigma appears as a simple sunken opening (Fig. 2AB). Marginal channel, located on valve face/mantle junction, narrow and internally occluded with hymenes (Fig. 3AA).

For morphometric description 40 valves were analyzed under LM, and 25 valves under SEM.

Type locality: Rapa Nui, cave interior, Coordinates: 27°6′42″S 109°24′14″W, collected on 30.04.1988 by Prof. Georg Zizka from Senckenberg Institute, Frankfurt am Main, Germany.

Holotype: holotype designated here: slide SZCZ20608 (Fig. 3H) in the collection of Andrzej Witkowski at the Institute of Marine and Environmental Sciences, University of Szczecin.

Isotype: slide number 2020/11 in the Diatom Collection of the Institute of Agricultural Sciences, Land Management and Environmental Protection at the University of Rzeszów.

Etymology: The species name refers to the island (being locus typicus) in the native language and to the indigenous people of the island itself.

Luticola moaiorum sp. nov. Peszek, M. Rybak, Witkowski & Lange-Bertalot (Fig. 4).

Figure 4 Luticola moaiorum sp. nov.

(A–N) LM images showing size diminution series. (H) Holotype specimen. (I) Isotype specimen. (O–Z) SEM images. (O–T) External valve views, note stigma position (arrow in Fig. T). (U) Close up of the specimen illustrated in Fig. P, illustrating the central area, with stigma covered by a mineral particle (arrow) and proximal raphe endings deflected to one side. (V) Close up of the valve apex with distal raphe ending and siliceous ridges on valve face/mantle conjunction evident (arrow). (W, X) Internal view of the entire valve. (Y) Close up of the specimen illustrated in Fig. X, showing the central area valve apex; note the marginal channel (arrow). (Z) Close up of the specimen illustrated in Fig. X, showing central area and stigma (arrow). Scale bar: (A–N, P) 10 µm, (O, Q–X) 5 µm, (Y) 4 µm, (Z) 1 µm.

Light microscopy: Valve outline symmetrical, valves elliptic-lanceolate with broadly rounded apices. Valve length 8.9–24.3 µm, width 4.3–7.9 µm. Axial area narrow and linear, central area bow-tie-shaped bordered by a solitary row of areolae. Transapical striae radiate becoming strongly radiate near apices (Figs. 4A–4N), 18–24 in 10 µm with 3–4 areola per stria (14–18 in 10 µm).

Scanning electron microscopy: Externally, raphe branches straight (Figs. 4O–4T), with proximal raphe endings distinctly bent to the side opposite stigma (Figs. 4O–4T and 4U), distal raphe endings also bent to the side opposite to stigma (Figs. 4O–4T and 4V). Areolae rounded to slightly elongated, radiate, 3–4 per stria (Figs. 4U and 4V), with a single row of areolae present on valve mantle (Figs. 4O, 4R and 4Q) Siliceous ridges present on valve face/mantle conjunction in almost half of observed specimens (Fig. 4V).

Internally, raphe simple and straight (Figs. 4W and 4X). Areolae covered by a single expansive hymenes forming continuous strips (Fig. 4Y). Stigma located in central area, about mid-way between the valve margin and the center, with a small C-shaped opening lipped structure (Figs. 4Y and 4Z). Marginal channel, located on valve face/mantle junction, narrow and occluded with hymenes (Fig. 4Y). On the side opposite the stigma the channel is wider and extends across the expanded central area (Figs. 4W, 4X and 4Y).

For morphometric description 40 valves were analyzed with LM, and 22 valves with SEM.

Type locality: Rapa Nui, cave interior, Coordinates: 27°6′42″S 109°24′14″W, collected on 30.04.1988 by Prof. Georg Zizka from Senckenberg Institute.

Holotype: holotype designated here: slide SZCZ20608 (Fig. 4H) in the collection of Andrzej Witkowski at the Institute of Marine and Environmental Sciences, University of Szczecin.

Isotype: slide number 2020/11 in the Diatom Collection of the Institute of Agricultural Sciences, Land Management and Environmental Protection at the University of Rzeszów.

Etymology: The name refers to the monolithic human figures on Rapa Nui, which are its most characteristic landscape feature.

Luticola ectorii Levkov, Metzeltin & Pavlov (Fig. 5).

Description: Valves rhombic to rhombic-elliptic, slightly asymmetric. Valve length 9.5–20.1 µm, width 5.2–7.0 µm, with 22–24 striae in 10 µm. Apices broadly rounded. Central area bow-tie shaped, bordered by single row of areolae (Figs. 5A–5J). In the SEM, the valve is ornamented by irregular depressions (Figs. 5L and 5M). The raphe is straight, with external proximal raphe ends deflected away from the isolated stigma, which is located in the central area (Figs. 5K and 5M). External distal raphe ends hooked to the side opposite the proximal ends and towards the stigma, continuing onto the valve mantle (Fig. 5N). Transapical striae composed of 4–5 rounded areolae, with a single row of areolae on valve mantle (Fig. 5N).

For morphometric description 15 valves were analyzed with LM, and 10 valves with SEM.

Taxonomic remarks: Specimens of the observed population had a smaller range of dimensions than those presented in the monograph of the genus (Levkov, Metzeltin & Pavlov, 2013). However they possess features typical of this species that allowed for its identification. These features include: the middle part being weakly, asymmetrically swollen, broadly rounded apices, slightly asymmetrical central area, stria and areola densities, and irregular depressions on the valve face.

Figure 5 Luticola ectorii Levkov, Metzeltin & A.Pavlov.

(A–J) LM images showing size diminution series. (K–N) SEM images of the valve illustrating the valve exterior, note the stigma in Fig. K (arrow) and proximal black arrow in (M) and distal raphe endings white arrow in (N) bent in one side. Scale bars: (A–J) 10 µm, (K–M) 5 µm, (N) 4 µm.

Ecology and associated diatom flora

In the sample from which the newly described Luticola species originate, abundant populations of other terrestial taxa were observed (Figs. 6 and 7). The most abundant of them were: Humidophila deceptioensis Kopalová, Zidarova & Van de Vijver (∼58%), Humidophila gallica (W.Smith) Lowe, Kociolek, Q.You, Q.Wang & Stepanek (∼10%), Humidophila cf. pantropica (Lange-Bertalot) Lowe, Kociolek, J.R.Johansen, Van de Vijver, Lange-Bertalot & Kopalová (∼16%) and Tryblionella debilis Arnott ex O’Meara (∼4%). Additionally, various terrestrial species such as Angusticopula chilensis (Grunow) Houk, Klee & H.Tanaka, Achnanthes tumescens A.R.Sherwood & R.L.Lowe, Halamphora montana (Krasske) Levkov, Humidophila contenta (Grunow) Lowe, Kociolek, J.R.Johansen, Van de Vijver, Lange-Bertalot & Kopalová, Humidophila sp., Geissleria ignota (Krasske) Lange-Bertalot & Metzeltin and two unidentified Orthoseira species were common in the studied sample. In the studied sample, other taxa identified included: Cavinula sp., Denticula subtilis Grunow, Fallacia insociabilis (Krasske) D.G.Mann, Ferocia ninae Van de Vijver & Houk, Halamphora normanii (Rabenhorst) Levkov, Mayamaea permitis (Hustedt) K.Bruder & Medlin, Navicula veneta Kützing, Nitzschia inconspicua Grunow, Nitzschia cf. microcephala Grunow, Nitzschia vitrea G.Norman, Pinnularia borealis Ehrenberg, Rhopalodia brebissoni Krammer, Sellaphora atomoides (Grunow) Wetzel & Van de Vijver, Sellaphora saugerresii (Desm.) Wetzel & D.G.Mann and Staurosirella pinnata (Ehrenberg) D.M.Williams & Round. These taxa appeared much less frequently, often in the form of single specimens and many of them had already been reported from the Rapa Nui Island by Cocquyt (1991).

Figure 6 Associeted diatom flora (LM images).

(A–D) Angusticopula chilensis (Grunow) Houk, Klee & H.Tanaka. (E–G) Ferocia ninae Van de Vijver & Houk. (H) Staurosirella pinnata (Ehrenberg) D.M.Williams & Round. (I–K) Achnanthes tumescens A.R.Sherwood & R.L.Lowe. (L–N) Navicula sp. (O–Q) Cavinula lapidosa (Krasske) Lange-Bertalot. (R–U) Geissleria ignota (Krasske) Lange-Bertalot & Metzeltin. (V–Y) Sellaphora atomoides (Grunow) Wetzel & Van de Vijver. (Z–AC) Sellaphora saugerresii (Desm.) Wetzel & D.G.Mann. (AD–AF) Halamphora montana (Krasske) Levkov. (AG–AJ) Halamphora normanii (Rabenhorst) Levkov. (AK–AN) Tryblionella debilis Arnott ex O’Meara. (AO, AP) Pinnularia borealis Ehrenberg. (AR) Nitzschia vitrea G.Norman. (AS–AU) Fallacia insociabilis (Krasske) D.G.Mann. (AV–AA) Humidophila sp. (AB–BG) Humidophila cf. pantropica (Lange-Bertalot) Lowe, Kociolek, J.R.Johansen, Van de Vijver, Lange-Bertalot & Kopalová. (BH, BI) Humidophila gallica (W.Smith) Lowe, Kociolek, Q.You, Q.Wang & Stepanek. (BJ–BP) Humidophila deceptioensis Kopalová, Zidarova & Van de Vijver. (BQ–BT) Humidophila contenta (Grunow) Lowe, Kociolek, J.R.Johansen, Van de Vijver, Lange-Bertalot & Kopalová. (BU–CX) Nitzschia cf. microcephala Grunow. (CY–CB) Nitzschia inconspicua Grunow. (CC–CE) Denticula subtilis Grunow. Scale bar: 10 µm.

Figure 7 SEM images for small-celled diatoms from the assemblage studied.

(A) Humidophila cf. pantropica (Lange-Bertalot) Lowe, Kociolek, J.R.Johansen, Van de Vijver, Lange-Bertalot & Kopalová. (B) Humidophila sp. (C) Humidophila gallica (W.Smith) Lowe, Kociolek, Q.You, Q.Wang & Stepanek. (D) Humidophila contenta (Grunow) Lowe, Kociolek, J.R.Johansen, Van de Vijver, Lange-Bertalot & Kopalová. (E, F) Humidophila deceptioensis Kopalová, Zidarova & Van de Vijver (E: external view, F: internal view). (G) Sellaphora saugerresii (Desm.) Wetzel & D.G.Mann. (G) Denticula subtilis Grunow. (I) Nitzschia cf. microcephala Grunow. (J) Nitzschia inconspicua Grunow. Scale bar: (A, B, G–J) 5 µm, (C) 3 µm, (D–F) 4 µm.

Discussion

Recent studies show that numerous species of Luticola can be found across a diverse range of habitats worldwide. As shown in literature most Luticola taxa have a distinct preference for wet limno-terrestrial environments including: edges of streams, wet soils or living on bryophytes (Round, Crawford & Mann, 1990; Kociolek et al., 2017).

Although a large number of Luticola species have been described as new to science in recent years, it was impossible to identify 3 of the 4 Rapa Nui Luticola populations we studied based on this currently available literature. Some previously-described Luticola species do show similarities with the newly described taxa. Included in this group are e.g., L. spinifera (W.Bock) L.Denys & W.H.De Smet, L. frequentissima Levkov, Metzeltin & A.Pavlov, L. puchalskiana Kochman-Kedziora, Zidarova, T.Noga, Olech & B.Van de Vijver and Luticola andina Levkov, Metzeltin & A.Pavlov (Levkov, Metzeltin & Pavlov, 2013; Kochman-Kedziora et al., 2020). However, none of the above species conformed with our new species in terms of size dimensions and/or ultrastructural characteristics.

Luticola georgzizkae sp. nov. possesses a combination of features unique within Luticola. So far only two of the established species in Luticola possess marginal spines –L. spinifera and L. lagerheimii (Cleve) D.G.Mann, but the shapes of their valves and their dimensions (Levkov, Metzeltin & Pavlov, 2013) easily separate them from L. georgzizkae. In addition, the two aforementioned taxa are characterized by a single morphology, whereas the newly described species has two morphotypes (spine-bearing and spineless forms). Moreover, in the case of L. spinifera and L. lagerheimii, the spines are positioned only on the edge of the valve and support formation of ribbon-like colonies. In the case of L. georgzizkae sp. nov. conjoined valves were not observed and the spines are also present directly on the surface of the valve. Denys & De Smet (1996) in studying Luticola spinifera suggested that spines present were capable of strong adhesion to the substratum. Colony formation might offer some compensation for the potential disadvantages of having small-sized frustules, such as the possibility of movement by wind. Colony formation also significantly reduces the area from which water may be quickly lost. Unfortunately growing in colonies limits the ability of individual cells to move, reducing the chances of finding more suitable environmental conditions in a constantly-changing terrestrial environment (Bock, 1963; Denys & De Smet, 1996). It seems that Luticola georgzizkae sp. nov. was able to combine both strategies, by creating forms with and without spines. In this study we were unable to observe colony formation, but this may be due to the low number of observations, or disconnection due to the sample cleaning process.

A similar phenomenon has also been observed in Humidophila gallica (W.Smith) Lowe, Kociolek, Q.You, Q.Wang & Stepanek (syn. Diadesmis gallica W.Smith). Mature valves of this species presented two morphotypes: one with a raphe system and no marginal spines, the other with marginal spines and without a raphe system (Cox, 2006). This same diatom species reduces the raphe length, following the development of colonies (Kociolek & Rhode, 1998). Also, in our study the spiny morphotype had a shorter raphe slit. It has also been suggested that the sealed raphe slit of an attached cell might minimize the risk of parasitism or infection (Cox, 2006). Which may mean that diatoms, at least those from terrestrial and aerophytic habitats, protect the cell content from external environment by reducing the raphe slit length or by sealing it off completely.

The second described species, Luticola rapanuiensis sp. nov. is most similar to L. frequentissima Levkov, Metzeltin & Pavlov. Both species possess thread-like depressions on the central raphe endings, but in L. frequentissima they are not conjoined with areolae from striae bordering the central area. Both taxa have similar size dimensions but valves of L. frequentissima are usually larger (5.2–15.9 µm long and 4.3–7.6 µm wide in L. rapanuiensis sp. nov. vs. 6.2–27.0 µm long and 4.0–9.0 µm wide in L. frequentissima) (Levkov, Metzeltin & Pavlov, 2013; Noga et al., 2017). Additionally, L. frequentissima has denser striae with higher areola number in an individual stria. Luticola frequentissima has 18–27 striae in 10 µm (Noga et al., 2017) composed of 4–6 areolae while L. rapanuiensis sp. nov. has 16–19 striae in 10 µm composed usually of 2–3 areola. Moreover, L. rapanuiensis sp. nov. has siliceous ridges over valve face/mantle conjunction which has not been observed in L. frequentissima (Levkov, Metzeltin & Pavlov, 2013).

Luticola moaiorum sp. nov. belongs to a group of small species that have elliptic-lanceolate valve outlines. From this group the most similar taxa are L. ipevii Van de Vijver & Levkov, L. puchalskiana Kochman-Kedziora, Zidarova, Noga, Olech & Van de Vijver and L. andina Levkov, Metzeltin & Pavlov. The newly described species can be easily distinguished from L. ipevi based on denser striae (18–24, vs. 14–18 in 10 µm) and mainly shorter and narrower valves. Another important characteristic is that L. ipevii has distal raphe fissures that continue onto the valve mantle (Levkov, Metzeltin & Pavlov, 2013). Also, large valves of L. ipevii show slightly separated, broadly rounded apices while L. moaiorum sp. nov. has a regularly elliptic-lanceolate valve shape across the size range. The basic feature that distinguishes Luticola moaiorum sp. nov. from L. puchalskiana is shape of central and distal raphe endings, which are simply bent to site opposite to stigma in first species, while in case of L. puchalskiana both raphe endings are hooked toward stigma-bearing side. The shape of the valves of both species is also different. In L. moaiorum sp. nov. the valves are elliptic-lanceolate while in L. puchalskiana they are rhombic-lanceolate (Kochman-Kedziora et al., 2020). Luticola andina differs from L. moaiorum sp. nov. by the shape of the central area which is elliptical to rectangular, not bow-tie shaped, and by a distinct axial area which is narrow in case of L. moaiorum sp. nov. Also newly described species show more radiate striae than L. andina (Levkov, Metzeltin & Pavlov, 2013).

It is interesting to note that while there are 4 species present in the sample from Rapa Nui, they are quite different from one another morphologically. For example, L. georgzizkae, L. rapanuiensis and L. moaiorum have external central and distal raphe ends that are curved in the same direction, though in the former the raphe ends are curved towards the stigma while in the latter two species they are directed away from the stigma. In L. ectorii the central and distal raphe ends are deflected in opposite directions. In three of the species, the internal structure of the stigma is similar to most Luticola species, a circular and lipped structure having an indistinct opening (e.g., Levkov, Metzeltin & Pavlov, 2013), while in L. rapanuiensis the internal stigma structure is simpler, with an opening in the center of the stigma. About the internal marginal channels, in L. georgzizkae they are prominent and tube-like, in L. rapanuiensis are barely resolvable, and in L. moaiorum there is a prominent extension of the channel across one side of the central area. This diversity of morphologies in the few species present on Rapa Nui might suggest these species arrived from different dispersal events, and are not a radiation from a single immigrant.

Diatom studies from Rapa Nui, though very few, do present some observations on Luticola species in the crater lakes and crater lake sediments. Firstly Cocquyt (1991) reported an occurrence of Luticola mutica (as Navicula mutica) in the crater lakes and this observation is documented with a line drawing. Subsequently, Dumont et al. (1998) published results of diatomological analyses of a sediment core from the crater lake and provided a taxa list of diatoms identified. This list includes mostly cosmopolitan taxa and, amongst them, three species of Luticola identified (as Navicula s.l.). These species included Navicula goeppertiana (Bleisch) H. L. Smith (=Luticola goeppertiana (Bleisch) D.G.Mann), Navicula goeppertiana var. monita (Hustedt) Lange-Bertalot (=Navicula monita Hustedt; Luticola monita (Hustedt) D.G.Mann) and Navicula mutica (=Luticola mutica). With the above taxa the number of Luticola taxa in Rapa Nui terrestrial and limnic waters increases to seven, three of which are new to science. It is worth comparing these figures with the closest continental mass which is the coast of South America and the Andes Mountains. Rumrich, Lange-Bertalot & Rumrich (2000) in their monograph identified eleven Luticola species in samples representing one of the Earth’s longest mountain ranges taken from the Feuer Land up to Panama. Among the reported Luticola species only two had been described as new to science and an additional seven were identified to the genus level only implying, they may potentially also be species new to science. Furthermore, of numerous cosmopolitan Luticola species, only L. goeppertiana was observed in the Andes. Also, when we analyze the Subantarctic Islands the figures in terms of Luticola species diversity are similar even though, as a rule, more extensive sampling has been done. This is fairly different from the case of Rapa Nui, where in one sample we have been able (thus far) to identify and describe three species new to science.

The identification of Luticola (Navicula) mutica from limnic waters of Rapa Nui may be perceived as problematic, since this taxon is considered a mud-dwelling, brackish water species (Levkov, Metzeltin & Pavlov, 2013). However, an argument in support of L. mutica occurring in freshwaters of Rapa Nui comes from species composition of the surface waters and of the core from the crater lake. In the species lists from both sites diatom taxa which are considered brackish water forms are included. Examples in this group are: Cyclotella meneghiniana, Navicula cf. phyllepta, Nitzschia cf. vidovichii or N. vitrea. We interpret this as an indication of strong winds which transfer an aerosol and sea salts from the sea coastal zone to the freshwater habitats.

Conclusions

Taking into consideration all described Luticola species, it seems that three new species described in this paper have a unique set of characters allowing their correct identification. So far, Luticola georgzizkae sp. nov., Luticola rapanuiensis sp. nov., and Luticola moaiorum sp. nov. are known only from Rapa Nui Island. Taking into account the fact that there is a high degree of endemism within Luticola (Kociolek et al., 2017), as well as a high degree of island isolation, it is very possible that the endemic species described from Rapa Nui do not exist elsewhere.

Supplemental Information

Data S1 Raw data (dimensions and number of striae) for all identified Luticola specimens

Click here for additional data file.

The authors express their gratitude to Professor Dr. Georg Zizka for providing the samples from Rapa Nui collected during his work on flowering plants.

Additional Information and Declarations

Competing Interests

Author Contributions

Data Availability

New Species Registration

The authors declare there are no competing interests.

Łukasz Peszek performed the experiments, analyzed the data, prepared figures and/or tables, authored or reviewed drafts of the paper, and approved the final draft.

Mateusz Rybak performed the experiments, analyzed the data, authored or reviewed drafts of the paper, and approved the final draft.

Horst Lange-Bertalot and John Patrick Kociolek analyzed the data, authored or reviewed drafts of the paper, and approved the final draft.

Andrzej Witkowski conceived and designed the experiments, performed the experiments, analyzed the data, authored or reviewed drafts of the paper, and approved the final draft.

The following information was supplied regarding data availability:

Raw data are available as a Supplemental File. Measurements for individual specimens are based on type material number SZCZ20608, stored in the collection of Andrzej Witkowski at the Institute of Marine and Environmental Sciences, University of Szczecin.

The following information was supplied regarding the registration of a newly described species:

Luticola georgzizkae

Luticola rapanuiensis

Luticola moaiorum.

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
