# Peer review of "Three new Luticola D.G.Mann (Bacillariophyta) species from Rapa Nui (Easter Island) found in terrestrial diatom assemblages dominated by widely distributed taxa"

_PeerJ, doi:10.7717/peerj.11142_

## Round 0.1 · original submission · Minor Revisions

The three reviewers coincided in that the paper is a nice contribution to the taxonomy of Luticola. They carefully read your article and made particular suggestions which you need to carefully consider, one by one and explain how you dealt with these suggestions in a rebuttal letter.

Reviewer 1 ·

Basic reporting

see additional notes under general comments.
-- Some editing of sentence structure, grammar, correction of typos will need to be completed. Otherwise, the writing is clear.
-- Sufficient and appropriate references are used. Refs – check for italics of genus and species names. (also spacing, use of periods after years, colon versus comma after journal volume, etc.).
--Manuscript goals were laid out.

Experimental design

The manuscript contains original primary research and contributes to knowledge on diatom taxonomy with respect to the genus Luticola. See notes below for how authors can strengthen their description of the research gap.
-- Authors provide appropriate detail and descriptions for the new species proposed.
-- Image plates are professionally laid out and conform to discipline standards.
-- Though the sample size is limited, the authors explain the origin of the samples and identify the limitations in the discussion.

1) Methods – Suggestions for additional details needed in the methods section.
a. Include a general map of the global location of the island and sampling site. Is there an image of the cave?
b. Sample collection: Where in the cave was the sample collected (floor, wall, etc). Was the sample analyzed from the soil associated with the plant collection, or was it collected with diatoms in mind (it appears as if it was associated with the plant collection). Provide additional detail here.
c. L135 – replace ‘community’ with ‘assemblage’ (community implies multiples trophic levels). Do a check throughout the manuscript.
d. Provide information relevant to new species descriptions. How many valves/frustules were measured under the LM for morphometric descriptions? How many under the SEM for ultrastructure?
e. Include the literature used for identifications and for the diatom terminology. Consider using parallel terminology for raphe ends: for example, you call distal raphe ends “distal”, but the “proximal” raphe ends “central”.

Validity of the findings

Results are valid. Data provided, include appropriate sample analysis and image documentation. Diatom repositories identified. Discussion is appropriate and links to research goals and is within the bounds of the results.

Additional comments

General comments
The authors use light and field emission microscope evidence from analysis of a diatom sample from a cave on the isolated island of Rap Nui (Easter Island) to describe and document three new species of Luticola (L. georgzizkae, L. rapanuiensis and L. moaiorum). These new taxa are compared to another Luticola taxon present in the sample, L. ectorii, and other Luticola species. The diatom diversity at this site, reported as low, contained other taxa common to aerophytic habitats. These descriptions and documentation of Luticola add to our understanding of this diatom genus and the diversity in soil-associated diatom floras. The remote nature of the island provides information relevant to diatom biogeography that the authors discuss.

The article will be of interest to readers of the journal Peer-J, especially those interested in diatom taxonomy, Luticola, biogeography. And cave, aerial and limno-terrestrial habitats. I enjoyed this manuscript.

2) Abstract:
a. L29: Add in a little more information about the type of samples/location within the cave (e.g. soil from a plant collection on the floor of the cave? Near the entrance, etc.). This will tell the reader a little more about the type of sample collected in the abstract before article download.

b. L36 – Include here the information the descriptions are based on (e.g. external and internal morphological features from light and field emission scanning electron microscopic analysis; versus genetic analysis)

3) Introduction.
a. The first part of the introduction provides a brief history of the study of diatoms on Rapa Nui. It would strengthen the case to flesh out this introductory paragraph to set the stage for the diversity documented to date, and any habitats that remain to be studied (inland areas??? statues? caves? Other? - are caves common in the inland or the coast?). Do you expect endemism or common taxa at these sites (why/why not….you hint at some thoughts/evidence here).
i. Of the freshwater floras examined to date, did the taxa documented contain new species? For example, L44-45 – How many of the 70 taxa were new to science? common?
ii. Were the diatoms from the core diverse? Any new to science?
iii. Are there other inland habitats, such as caves or the stone statues where the diatom flora remains underexplored? Given the remote nature and history of this island, the diatom flora of the sites studied to date, do you expect these habitats to contain other new taxa or mostly common taxa?
iv. …why this location…why now.. (It’s an interesting habitat and a unique location… …)

b. L50 – start a new paragraph here. You will need to add a connector set of sentences to go from the introduction of the island to the introduction of Luticola.

c. Carefully consider the use of the term aerophytic to confirm this is the correct term. There are a number of terms out there that describe different types of aerial to terrestrial environments, for example depending on the moisture source. There is not enough information here to determine if aerophytic is the right term. You associate the term soil in parentheses beside the word and reference that samples were collected from a cave. Take a short paragraph in the methods or intro to shore up this definition and to clear up how you use the term throughout.

4) Results.
a. New species descriptions are consistent and appropriate (watch for typos). See some comments below for strengthening the connection between the descriptions and the image plates. It is great to have a nice LM size series for each new taxon, and to have multiple SEM images for each taxon! Thank you for taking the time to do that. Plates are well organized and laid out. Consider running a temporary line horizontally across the middle to line up the central area of each specimen (I think you will find a few will get reordered to create the size diminution series).

b. Summarize the measurements/features of the Luticola species in a table for future comparison or reference. You could also consider adding in measurements for some of the similar Luticola species. For example, like the tables in Pavlov et al. (2009) on Luticola grupcei, or Levko, Metzeltin, Pavlov (2013)

c. Given the unique location and the expanding studies of aerophyllic/soil-associated habitats, it would absolutely be fabulous to have an additional plate with a representative image of the other taxa.

5) Results: Figures/Captions.
a. Overall, I would add a little more detail to the caption to identify features included in the species description. Add in additional arrows.

b. Fig.1.: Add in a little more detail. A-N – size diminution series (relevant to all captions). Indicate which is the image of the holotype and lectotype.

You indicate the “H” is a spineless (implied SEM given sequence) – but H is an LM. Reword or re-order phrasing.
(Z) arrow stated in Fig. caption – none on the image (add in)

c. Fig. 2. Add in a little detail. For example, what do the external views highlight or show? What about the girdle or internal view? (Q-w) did you mean valve? Or vale? Arrow only on U and V – either add arrows or reword to be clear. Identify/include the holotype and lectotype images.

d. Fig. 3 V – it mentions that “V” shows the stigma (looks to be covered with debris)…reference a different image for this feature. Z – mentions arrow – add in the arrow in the image. Identify in the caption and in the image the holotype and lectotype.

e. Luticola rapanuiensis. L197 did you mean “within”?
I’m not sure undulate is the right word to capture the raphe shape…or maybe broadly undulate or…? (or maybe it is…maybe revisit). Where you describe the irregular shallow grooves by the proximal raphe ends, do you mean to reference these as ghost areolae? You label them as such in the figure caption, but not necessarily in the description. Be clear and consistent. You may also want to indicate in the caption that the valve in image V is tilted to skew the view of the valve face/ highlight its topography.

6) Discussion -

a) The discussion points seem relevant and provide appropriate comparisons with similar taxa.

b) The authors may consider making a point on how the source of the sample (e.g. a botanical sample) may provide other unique opportunities to study diatoms from unique or understudied environments. Any plans to continue the study on the diatom flora of Rapa Nui?

7) Minor/other:
a) There are lots of inconsistencies around the italics of the parentheses of the in-text citations (some are italicized, others are not, or one is and one is not). Do a careful edit of these.

b) L44 missing a word after microscopy-based? (or sentence needs an edit). -- Overall, there are numerous typos or minor grammar corrections needed throughout the manuscript. For example., L393 “not exits on” (“do not exist at”); rhomic-elliptic (did you mean rhombic-elliptic); “Levkow et. al., 2013, refs” – should be Levkov; “Rap” Nui versus “Rapa Nui”.

c) You flip between the use of Rapa Nui and Easter Island. Consider if you want to present the location consistently one way or the other throughout or if you want to use both. For example, in the discussion on biogeography you use Easter Island, but then for the conclusion, flip back to Rapa Nui. (I might lean toward using the original name..like you do in the title and abstract)

Reviewer 2 ·

Basic reporting

Dear Editor,
The submitted manuscript reports diatom findings from a single sample taken from cave on the isolated island Rapa Nui. Among other aerophilic genera, four Luticola species were found in the sample and three out of them are described in the manuscript as new to science.

Experimental design

A single samples was examined for the diatom species composition under the LM and SEM microscopy which is basic and common treatment for the diatom research. Presented species are well documented and described in the text. It is clear, that authors have experience in the taxonomic work and are familiar with the genus Luticola. The important comparison with other similar species is included.

Validity of the findings

The genus Luticola has been recently very popular diatom genus as model organism to study the species distribution and its biogeography. Many new species have been described all over the world during last decides and we can see a high number of endemic species presented only in specific areas. Specifically, in this study, the geographical isolation is so high that it is not surprising that species new to science have been found there. It is very species rich genus accommodating around 240 species these days. The genus has received many attention after its generic establishment, recently mainly withing the new monograph written by Levkov et all and work done at south-Asia and Antarctica. Many times we can hear that those regions are the diversity hot spots, however the question is, if that is not influenced just by under-sampling of other localities worldwide. So from that point of view I see very important and I welcome very much any record from other parts of the world presenting the species composition and distribution of the Luticola species.

Additional comments

It is well written manuscript but what would be excellent is if authors treating (not only) this genus would consider including also molecular data of the species as it is the key to uncover the real diversity of the genus Luticola.

Reviewer 3 ·

Basic reporting

For the Introduction, I recommend checking the articles “Bustos S., Morales M. R. & Maidana N. I. 2017. Diversidad del género Luticola (Bacillariophyceae) en sedimentos holocenicos de la Puna Jujeña, Argentina. Boletín de la Sociedad Argentina de Botánica 52: 13–25.”, “Straube A., Tremarin P. I. & Ludwig T. A. V. 2017. Species of Luticola D.G. Mann (Bacillariophyceae) in the Atlantic Forest rivers from southern Brazil. Diatom Research 32: 417–437.”, and “DA SILVA-LEHMKUHL A. M., Ludwig T. A. V., Tremarin P. I. & Bicudo D. C. 2019. On Luticola Mann (Bacillariophyceae) in Southeastern Brazil: Taxonomy, ecology and description of two new species. Phytotaxa 402: 165–186.” in which describe new species of Luticola in the Andes and South America rainforest.

I considered that the background about vascular plants is a little bit extensive, maybe you can shorten this.

If you want you can add a map to highlight the isolation of the island.

I think when you talk about a structure is necessary to add references to the figures where the structure can be observed.

Do you have another picture of the internal view of Luticola rapanuiensis? I am not sure about the stigma internal opening, I saw a mark around the opening, maybe is a scar of the stigma structure.

About the stigma internal opening in Luticola georgzizkae and Luticola moaiorum I think that in both species is a circular and lipped structure without a distinct opening that in the pictures is broken, basing me that in the Luticola moaiorum pictures X and Y the “opening” have different shapes.

Experimental design

No comment

Validity of the findings

No comment

Additional comments

I consider that the article is well structured and presents new relevant information to advance the understanding of the diversity and distribution of the Luticola genus. The species are well defined with strong characters that clearly differentiate them from those present in the literature.
I think the article only needs a few minor corrections, please check the pdf file with my comments.

Annotated reviews are not available for download in order to protect the identity of reviewers who chose to remain anonymous.

---

## Round 0.2 · accepted · Accept

I appreciate your effort in dealing with suggestions made by the three reviewers, they improved the manuscript.